# BiToD: A Bilingual Multi-Domain Dataset For Task-Oriented Dialogue Modeling

**Zhaojiang Lin**[1]*, **Andrea Madotto**[1]*, **Genta Indra Winata**[1], **Peng Xu**[1],
**Feijun Jiang**[2], **Yuxiang Hu**[2], **Chen Shi**[2], **Pascale Fung**[1]
[1]Center for Artificial Intelligence Research (CAiRE)
[1]The Hong Kong University of Science and Technology
[2]Alibaba Group
{zlinao, amadotto, giwinata, pxuab}@connect.ust.hk

## Abstract

Task-oriented dialogue (ToD) benchmarks provide an important avenue to measure progress and develop better conversational agents. However, existing datasets for end-to-end ToD modeling are limited to a single language, hindering the development of robust end-to-end ToD systems for multilingual countries and regions. Here we introduce BiToD[2], the first bilingual multi-domain dataset for end-to-end task-oriented dialogue modeling. BiToD contains over 7k multi-domain dialogues (144k utterances) with a large and realistic bilingual knowledge base. It serves as an effective benchmark for evaluating bilingual ToD systems and cross-lingual transfer learning approaches. We provide state-of-the-art baselines under three evaluation settings (monolingual, bilingual, and cross-lingual). The analysis of our baselines in different settings highlights 1) the effectiveness of training a bilingual ToD system compared to two independent monolingual ToD systems, and 2) the potential of leveraging a bilingual knowledge base and cross-lingual transfer learning to improve the system performance under low resource conditions.

## 1 Introduction

Task-oriented dialogue (ToD) systems are designed to assist humans in performing daily activities, such as ticket booking, travel planning, and online shopping. These systems are the core modules of virtual assistants (e.g., Apple Siri and Amazon Alexa), and they provide natural language interfaces for online services [1]. Recently, there has been growing interest in developing deep learning-based end-to-end ToD systems [2, 3, 4, 5, 6, 7, 8, 9, 10, 11, 12, 13, 14, 15, 16] because they can handle complex dialogue patterns with minimal hand-crafted rules. To advance the existing state-of-the-art, large-scale datasets [17, 1, 16] have been proposed for training and evaluating such data-driven systems.

However, existing datasets for end-to-end ToD modelling are limited to a single language, such as English [17, 18], or Chinese [19, 20]. The absence of bilingual or multilingual datasets not only limits the research on cross-lingual transfer learning [21] but also hinders the development of robust end-to-end ToD systems for multilingual countries and regions.

To tackle the challenge mentioned above, we introduce BiToD, a bilingual multi-domain dataset for task-oriented dialogue modelling. BiToD has 7,232 bilingual dialogues (in English and Chinese), spanning seven services within five domains, where each dialogue is annotated with dialogue states, speech-acts, and service API calls. Therefore, BiToD can be used for building both end-to-end ToD

---

* Equal contribution
[2]Data and code are available in `https://github.com/HLTCHKUST/BiToD`.

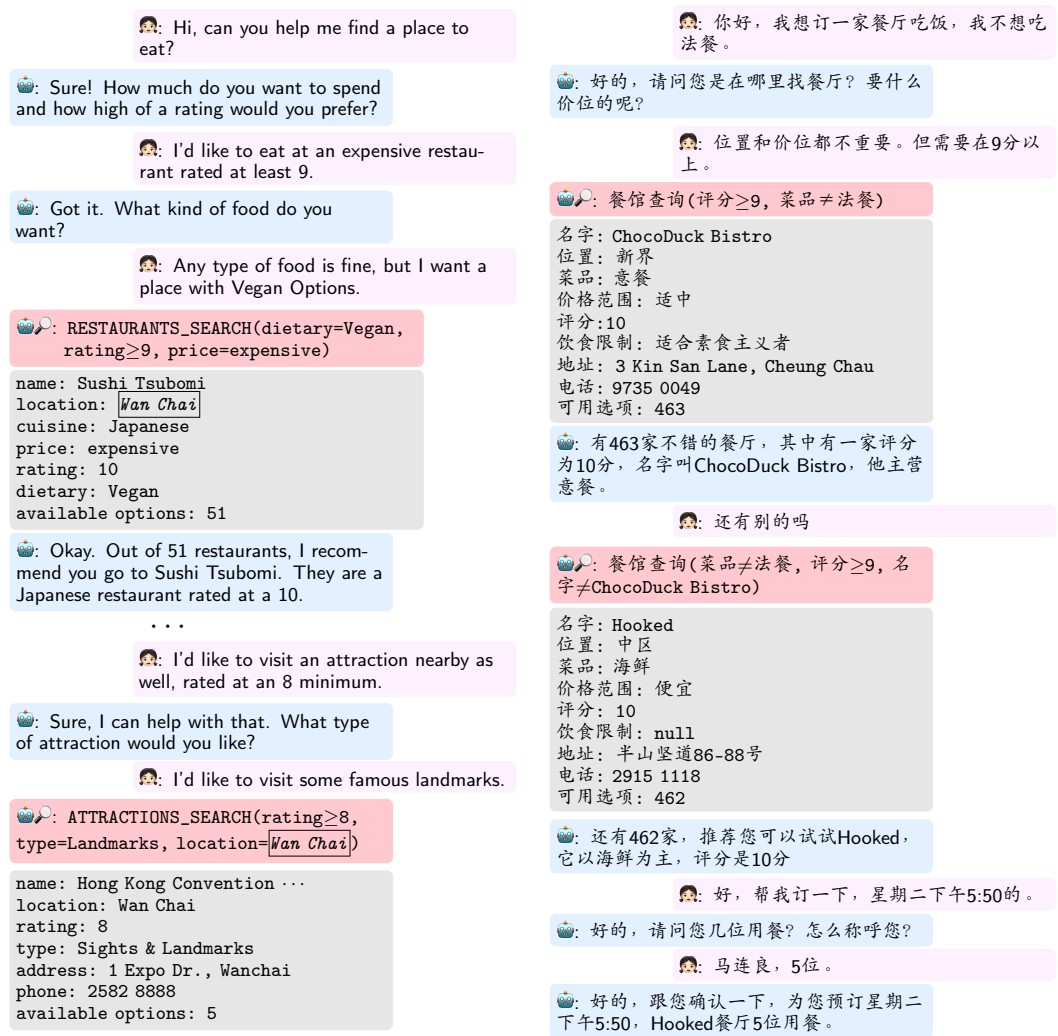

Figure 1: Dialogue examples in English (on the left) and Chinese (on the right) from the 𝔹𝕚𝕋𝕠𝔻 dataset (best viewed in colour). The user (in pink) asks for restaurant and attraction information. At the same time, the system (in blue) responds to the user utterance by calling APIs (in red) when needed and by incorporating the API results (in grey).

systems and dialogue sub-modules (e.g., Dialogue State Tracking). We propose three evaluation settings: 1) *monolingual*, in which the models are trained and tested on either English or Chinese data, 2) *bilingual*, where the models are trained with bilingual data and tested with English and Chinese dialogues simultaneously, and 3) *cross-lingual*, where the models are first trained with the source language and then tested in a few-shot setting in the target language.

The contribution of this work is three-fold. 1) We propose the first bilingual dataset (𝔹𝕚𝕋𝕠𝔻) with a total of 7, 232 dialogues for *end-to-end* ToD modeling. 𝔹𝕚𝕋𝕠𝔻 serves as an effective benchmark for evaluating bilingual ToD systems and cross-lingual transfer learning approaches. 2) We provide novel baselines under the three evaluation settings, i.e., monolingual, bilingual, and cross-lingual. 3) We show the effectiveness of training a bilingual ToD system compared to two independent monolingual ToD systems as well as the potential of leveraging a bilingual knowledge base and cross-lingual transfer learning to improve the system performance under low resource condition.

The paper is organized as follows: We next describe the 𝔹𝕚𝕋𝕠𝔻 data collection methods in Section 2. We then describe our proposed tasks in section 3. Section 4 introducew our baselines, and we finally present and discuss results in Section 5.

## 2 𝔹𝕚𝕋𝕠𝔻 **Dataset**

In this paper, we focus on one-to-one conversations that only involve two speakers (e.g., a user $U$ and a system $S$). A dialogue is a sequence of *utterances* ($U_1$, $S_1$, $U_2$, $S_2$, and so on), each a single contribution from one speaker to the dialogue [22]. In ToD, there can be additional API calls in system turn when the user requests the system to search about a certain information. In addition to aforementioned concept, there are several frequently used terms in ToD. *Domains* are the topics of the current conversation, for example, restaurant domain is about restaurant reservation and metro domain is about taking a metro to somewhere. *Multi-domain* dialogues are referred as dialogues that involves more than one domain. As shown in Figure 1, the English dialogue involves both restaurant and attraction domains.

𝔹𝕚𝕋𝕠𝔻 is designed to develop virtual assistants in multilingual cities, regions, or countries (e.g., Singapore, Hong Kong, India, Switzerland, etc.). For the 𝔹𝕚𝕋𝕠𝔻 data collection, we chose Hong Kong since it is home to plenty of attractions, restaurants and more, and is one of the most visited cities globally, especially by English and Chinese speakers. In 𝔹𝕚𝕋𝕠𝔻, most of Chinese dialogues are collected in simplified Chinese by native mandarin speakers (due to lack of Cantonese annotators). The hiring process is available in Appendix. This section describes the knowledge base construction and provides detailed descriptions of the dialogue collection.

### 2.1 Knowledge Base Collection

We collect publicly available Hong Kong tourism information from the Web, to create a knowledge base that includes 98 metro stations, 305 attractions, 699 hotels, and 1,218 restaurants. For the weather domain, we synthetically generate the weather information on different dates. Then, we implement seven service APIs (`Restaurant_Searching`, `Restaurant_Booking`, `Hotel_Searching`, `Hotel_Booking`, `Attraction_Searching`, `MTR_info`, `Weather_info`) to query our knowledge base. The knowledge base statistics are shown in Appendix. Although we aim to collect a fully parallel knowledge base, we observe that some items do not include bilingual information. For example, several traditional Cantonese restaurants do not have English names, and similarly, some restaurants do not provide addresses in Chinese. This lack of parallel information reflects the real-world challenges that databases are often incomplete and noisy.

### 2.2 Dialogue Data Collection

The dialogues are collected through a four-phase pipeline, as shown in Figure 2. We first design a schema, as a flowchart, for each service API, to specify the possible API queries and expected system actions after the API call. Then, user goals are sampled from the knowledge base according to the pre-defined schemas. Based on the user goals, the dialogue simulator interacts with the APIs to generate dialogue outlines. Finally, the dialogue outlines are converted into natural conversations through crowdsourcing. Our data collection methodology extends the Machine-to-Machine (M2M) approaches [23, 1] to bilingual settings to minimize the annotation overhead (time and cost).

**Schemas and APIs.** The dialogue schema shown as a flowchart (`Restaurant_Searching`) in Figure 2.a specifies the input and output options of the API and the desired system behaviours. To elaborate, the user searches a restaurant by name, location, cuisine, etc. Then the system calls the API and informs the user of the restaurant name and other requested information. If the user is not satisfied with the search results, the system continues searching and provides other options. To ensure the provided services are realistic, we impose a few restrictions, as in [1]. Firstly, each API has a list of required slots, and the system is not allowed to hit the API without specifying values for these slots. For example, the system needs to obtain departure and destination locations before calling the metro-info API. Secondly, the system must confirm the booking information with the user before making any reservations (e.g., restaurant booking).

**User Goals.** A user goal consists of a list of intents and a set of constraints under each intent. Figure 2.b shows a single domain (intent) example where the user's intent is *Restaurant_Search*. A constraint is defined with a triple (slot, relation, value) (e.g., (`Rating, at_least, 4`)). Different from previous work, which defined user constraints as slot-value pairs, we impose slot-value relations [18] (listed in Figure 3.b) to promote more diverse user goals. To generate a user goal, we first sample

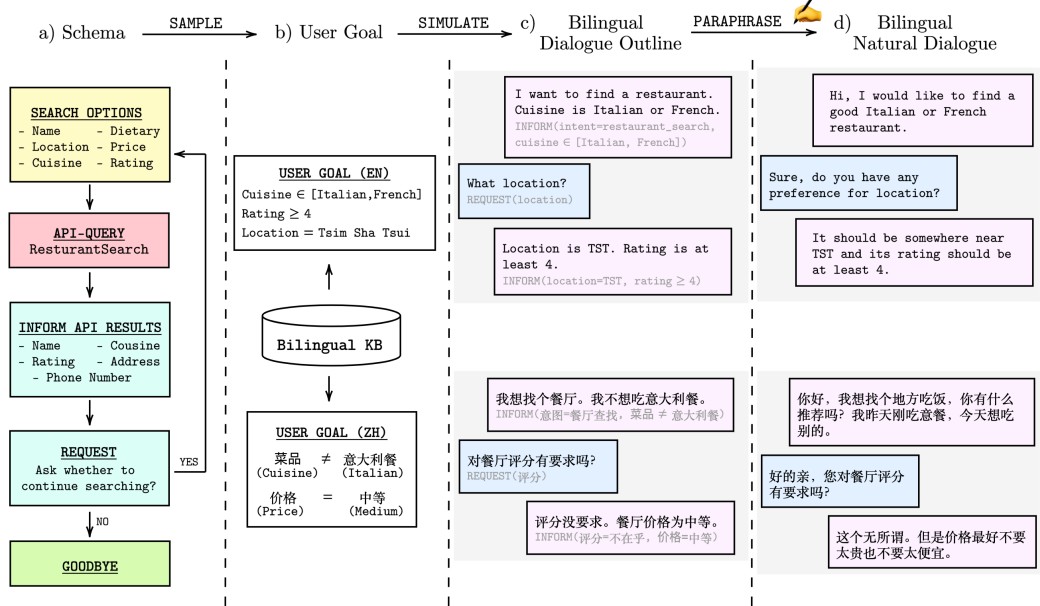

Figure 2: Illustration of the bilingual dialogues collection pipeline: a) Design a schema for each service API; b) Sample user goals from the bilingual knowledge base (KB) according to schema; c) Based on one of the user goals, the dialogue simulator generates the dialogue outlines while interacting with the APIs; d) Convert the dialogue outlines to natural conversations via crowdsourcing. Note that English and Chinese user goals are sampled independently.

a list of intents. We randomly sample a set of slot-relation-value combinations from the bilingual knowledge base for each intent, which includes non-existent combinations to create unsatisfiable user requests. In multi-domain scenarios, we set a certain probability to share the same values for some of the cross-domain slots (e.g., date and location) to make the transition among domains smooth. For example, users might want to book restaurants and hotels on the same date or take the metro from the location of the booked restaurant to their hotel. Note that the user goals for English and Chinese are sampled independently, as the real-world customer service conversations are often unparalleled.

**Dialogue Outline Generation.** Dialogue outlines are generated by a bilingual dialogue simulator that accepts user goals in both languages as inputs. The dialogue simulator consists of a user agent and a system agent. Both agents interact with each other using a finite set of actions specified by speech acts over a probabilistic automaton designed to capture varied dialogue trajectories [1]. Each speech act takes a slot or slot-relation-value triple as an argument. When the conversation starts, the user agent is assigned a goal, while the system agent is initialized with a set of requests related to the services. During the conversation, the user informs constraints according to the user goal, and the system responds to the user queries while interacting with the service APIs. For some services, the system needs to request all the required slots before querying the APIs. After the API call, the system either informs the search result or searches for other options until the user intents are fulfilled. Following [1], we also augment the value entities during the dialogue outlines generation process, e.g., *Tsim Sha Tsui* can be replaced with its abbreviation *TST*, as shown in Figure 2.c. After the user goal is fulfilled by a series of user and system actions, we convert all the actions into natural language using templates. In this phase, we obtain the dialogue states annotations and speech acts automatically for both the user and system sides.

**Dialogue Paraphrase.** The dialogue outlines are converted to natural dialogues via crowdsourcing. Appendix Figure 1 and 2 show the interface for Chinese and English paraphrasing, where workers see the full dialogue and rewrite the dialogue turn by turn. Before the task, workers are asked to read

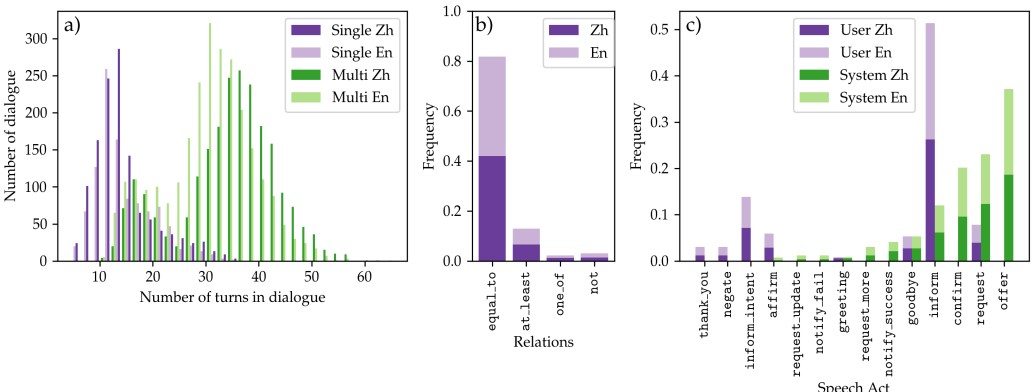

Figure 3: Data statistic of 𝔹𝕚𝕋𝕠𝔻: a) dialogue distribution of lengths of single and multi-domain dialogues, b) distribution of different relation types, and c) distribution of speech acts of users and systems.

the instructions, shown in Appendix Figure 3. In the instructions, we specify that the paraphrased dialogue should retain the same meaning as the dialogue outline but sound like a real conversation between a user and a professional assistant. The user utterances are expected to be creative and diverse, while the system utterances are expected to be formal and correct. To ensure all the essential information is presented in the new dialogue, we highlight all the entities with bold text. In the user utterances, the highlighted entities are allowed to be paraphrased without losing their original meaning; e.g., *"The restaurant should provide **Vegan Options"*** is allowed to be rewritten as *"I would like to find a **vegan-friendly** restaurant"*. In contrast, all the entities in the system utterances are required to be unchanged.

**Quality Verification.** After the dialogue paraphrasing, workers are asked to read through the new dialogue and answer the following questions, as in [23]: 1) *Does it seem like a conversation between a user that sounds like you and an assistant that sounds formal?* 2) *Does it have the same meaning as the original conversation, while still making sense on its own?* The first question is for examining whether the new conversation is realistic, and the second question is for verifying whether the dialogue outline and the paraphrased dialogue are valid. Given the two answer options: 1) *Yes*, 2) *No, but I cannot make it better*, 97.56% of annotators chose the first option for the first question and 98.89% of them chose the first option for the second question. After the dialogue paraphrasing, we randomly sampled around 3000 dialogues, and asked different sets of annotators to check the quality paraphrased dialogues (in terms of naturalness of the language and dialogue flow). We found around 103 (3.43%) low quality dialogues, and all of them have been annotated.

## 2.3 Dataset Statistics

We collected 7,232 dialogues with 144,798 utterances, in which 3,689 dialogues are in English, and 3,543 dialogues are in Chinese. We split the data into 80% training, 8% validation, and 12% testing, resulting in 5,787 training dialogues, 542 validation dialogues, and 902 testing dialogues. In Figure 3 we show the main data statistics of the BiToD corpus. As shown in Figure 3.a, the lengths of the dialogues vary from 10 turns to more than 50 turns. Multi-domain dialogues, in both English and Chinese, have many more turns compared to single-domains. The most used relation in user goals is `equal_to` (Figure 3.b), and the most common speech-acts (Figure 3.c) for users and systems are `inform` and `offer`, respectively. Finally, in Table 2 in the Appendix, we list all the informable and requestable slots per domain.

## 2.4 Dataset Features

Table 1 shows the comparison of the 𝔹𝕚𝕋𝕠𝔻 training set to previous ToD datasets. Prior work for end-to-end ToD modelling only focuses on a single language. Our 𝔹𝕚𝕋𝕠𝔻 is the first ***bilingual*** ToD

|  | **MultiWoZ** | **FRAMES** | **TM-1** | **SGD** | **STAR** | **RiSAWOZ** | **CrossWoz** | **BiToD** |
|---|---|---|---|---|---|---|---|---|
| *Language(s)* | EN | EN | EN | EN | EN | ZH | ZH | EN, ZH |
| *Number of dialogues* | 8,438 | 1,369 | 13,215 | 16,142 | 5,820 | 10,000 | 5,012 | 5,787 |
| *Number of domains* | 7 | 1 | 6 | 16 | 13 | 12 | 5 | 5 |
| *Number of APIs* | 7 | 1 | 6 | 45 | 24 | 12 | 5 | 7 |
| *Total number of turns* | 115,434 | 19,986 | 274,647 | 329,964 | 127,833 | 134,580 | 84,692 | 115,638 |
| *Average turns / dialogues* | 13.46 | 14.6 | 21.99 | 20.44 | 21.71 | 13.5 | 16.9 | 19.98 |
| *Slots* | 25 | 61 | - | 214 | - | 159 | 72 | 68* |
| *Values* | 4,510 | 3,871 | - | 14,139 | - | 4,061 | 7,871 | 8,206* |
| *Deterministic API* | ✗ | ✗ | ✗ | ✗ | ✗ | ✗ | ✗ | ✓ |
| *Complex User Goal* | ✗ | ✗ | ✗ | ✗ | ✓ | ✗ | ✗ | ✓ |
| *Mixed-Language Context* | ✗ | ✗ | ✗ | ✗ | ✗ | ✗ | ✗ | ✓ |
| *Provided KB* | ✓ | ✗ | ✗ | ✗ | ✓ | ✓ | ✓ | ✓ |

Table 1: Comparison of BiToD to previous ToD datasets. The numbers are provided for the ***training set*** except for FRAMES and STAR. *We consider entities in different language as different slots and values.

corpus with comparable data size. In addition to its bilingualism, 𝔹𝕚𝕋𝕠𝔻 also provides the following unique features:

**Deterministic API.** Given an API query for recommendation services (e.g., restaurant searching and hotel searching), there is typically more than one matched item. Previous works [17, 19] randomly sampled one or two items as API results and returned them to users. However, in real-world applications, the system should recommend items according to certain criteria (e.g., user rating). Moreover, the randomness of the API also increases the difficulty of evaluating the models. Indeed, the evaluation metrics in [17, 19] rely on delexicalized response templates, which are not compatible with knowledge-grounded generation approaches [10, 12]. To address these issues, we implement deterministic APIs by ranking the matched items according to user ratings.

**Complex User Goal.** To simulate more diverse user goals, we impose different relations for slot-value pairs. For example, in the restaurant searching scenarios, a user might want to eat Chinese food (`cuisine, equal_to, Chinese`), or do not want Chinese food (`cuisine, not, Chinese`). Figure 3.b shows the distribution of different relations in user goals.

**Mixed-Language Context.** Our corpus contains code-switching utterances as some of the items in the knowledge base have mixed-language information. In the example in Figure 1.b, the system first recommends a restaurant called *ChocoDuck Bistro* and the user asks for other options. Then the system searches other restaurants with an additional constraint (`restaurant_name, not, ChocoDuck Bistro`). In this example, both restaurants only have English names, which is a common phenomenon in multilingual regions like Hong Kong. Thus, ToD systems need to handle the mixed-language context to make correct API calls. Note that the code-switching utterances in our dataset are mostly restricted to the interchange between Chinese and English name entities, such as hotel names, attraction names.

**Cross-API Entity Carry-Over** Our corpus includes scenarios where the value of a slot is not presented in the conversation, and the system needs to carry over values from previous API results. In the example in Figure 1.a, the user first finds and books a restaurant without specifying the location; then she (🧑) wants an attraction nearby the restaurant. In this case, the system needs to infer the attraction location (𝑊𝑎𝑛 𝐶ℎ𝑎𝑖) from the restaurant search result.

## 3 Tasks & Evaluations

### 3.1 Dialogue State Tracking

Dialogue state tracking (DST), an essential task for ToD modelling, tracks the users' requirements over multi-turn conversations. DST labels provide sufficient information for a ToD system to issue APIs and carry out dialogue policies. In this work, we formulate a dialogue state as a set of slot-relation-value triples. We use Joint Goal Accuracy (**JGA**) to evaluate the performance of the DST.

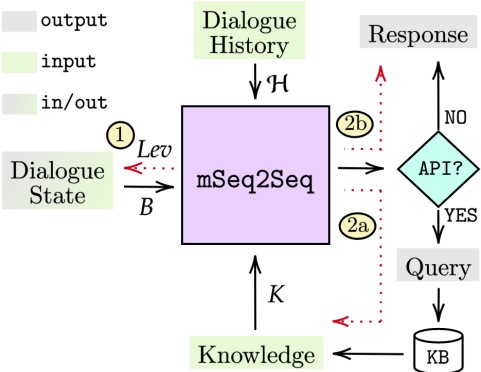

Figure 4: Model response generation workflow. Given the dialogue history $\mathcal{H}$, the knowledge $K$ (which can also be empty), and the dialogue state $B$, the mSeq2Seq 1) updates the dialogue state by generating $Lev$, and 2) generate a textual output and checks if it is an API or a Response. If the output is an API, (2a) the system queries the KB and updates the knowledge $K$; otherwise (2b) the Response is shown to the user. See this GIF for more details.

The model outputs are correct when all of the predicted slot-relation-value triples exactly match the oracle triples.

## 3.2 End-to-End Task Completion

A user's requests are fulfilled when the dialogue system makes correct API calls and correctly displays the requested information. We use the following automatic metrics to evaluate the performance of end-to-end task completion: 1) Task Success Rate (**TSR**): whether the system provides the correct entity and answers all the requested information of a given task, 2) Dialogue Success Rate (**DSR**): whether the system completes all the tasks in the dialogue, 3) API Call Accuracy (**API**$_{Acc}$): whether the system generates a correct API call, and 4) **BLEU** [24]: measuring the fluency of the generated response.

## 3.3 Evaluation Settings

**Monolingual.** Under the monolingual setting, models are trained and tested on either English or Chinese dialogues.

**Bilingual.** Under the bilingual setting, models are trained on bilingual dialogues (full training set), and in the testing phase, the trained models are expected to handle dialogues in both languages simultaneously without any language identifiers.

**Cross-lingual.** This setting simulates the condition of lacking data in a certain language, and we study how to transfer the knowledge from a high resource language to a low resource language. Models have full access to the source language in this setting but limited access to the target language (10%).

## 4 Proposed Baselines

Our proposed baselines are based on the recent state-of-the-art end-to-end ToD modeling approach *MinTL* [15] and cross-lingual transfer approach MTL [25]. We report the hyper-parameters and training details in the Appendix.

**Notations.** We define a dialogue $\mathcal{D} = \{U_1, S_1, \ldots, U_T, S_T\}$ as an alternating set of utterances from user and systems. At turn $t$, we denote a dialogue history as $\mathcal{H}_t = \{U_{t-w}, S_{t-w}, \ldots, S_{t-1}, U_t\}$, where $w$ is the context window size. We denote the dialogue state and knowledge state at turn $t$ as $B_t$ and $K_t$, respectively.

## 4.1 ToD Modeling

Figure 4 describes the workflow of our baseline model. We initialize the dialogue state $B_0$ and knowledge state $K_0$ as empty strings. At turn $t$, the input of our model is the current dialogue history $H_t$, previous dialogue state $B_{t-1}$ and knowledge state $K_{t-1}$. Similar to the text-to-text

transfer learning approach [26], we add a prompt $P_B = "TrackDialogueState :"$ to indicate the generation task. Then, a multilingual sequence-to-sequence (mSeq2Seq) model takes the flattened input sequence and outputs the *Levenshtein Belief Spans* ($Lev_t$) [15]:

$$(1): \text{Lev}_t = \text{mSeq2Seq}(P_B, \mathcal{H}_t, B_{t-1}, K_{t-1}).$$

The $Lev_t$ is a text span that contains the information for updating the dialogue state from $B_{t-1}$ to $B_t$. The updated dialogue state $B_t$ and a response generation prompt, $P_R = "Response :"$, are used as input. Then, the model will either generate an API name (2a) when an API call is needed at the current turn, or a plain text response directly returned to the user (2b). If the model generates an API name, it is

$$(2a): \text{API} = \text{mSeq2Seq}(P_R, \mathcal{H}_t, B_t, K_{t-1}),$$

the system will query the API with the constraints in the dialogue state and update the knowledge state $K_{t-1} \to K_t$. The updated knowledge state and API name are incorporated into the model to generate the next turn response generation.

$$(2b): \text{Response} = \text{mSeq2Seq}(P_R, \mathcal{H}_t, B_t, K_t, \text{API}).$$

All the aforementioned generation process are based on a single mSeq2Seq, and we initialized our model with two pre-trained models, **mT5** [26] and **mBART** [27].

### 4.2 Cross-lingual Transfer

Based on the modelling strategy mentioned above, we propose three baselines for the cross-lingual setting.

**mSeq2seq.** Directly finetune the pre-trained mSeq2seq models like mBART and mT5 on the 10% dialogue data in the target language.

**Cross-lingual Pre-training (CPT).** First, pre-train the mBART and mT5 models on the source language, then finetune the models on the 10% target language data.

**Mixed-Language Pre-training (MLT).** To leverage the fact that our knowledge base contains the bilingual parallel information for most of the entities, we replace the entities in the source language data (both input sequence and output sequence) with their target language counterpart in our parallel knowledge base to generate the mixed-language training data. We first pre-train the mSeq2seq models with the generated mixed-language data, then finetune the models on the 10% target language data.

**Translate Train (TT).** We study applying the state-of-the-art pre-trained neural machine translation model M2M-100 [28] to our cross-lingual transfer setting. We first translate the dataset of source language to target language, then train the model on the translated dataset. Finally, we finetune the trained model with the 10% target language data. Note that this approach can be applied only when we have additional resource to train a machine translation model from the source language to the target language.

## 5  Results & Discussion

The main results for DST and end-to-end task completion are reported in Table 2. Note that the $API_{Acc}$ is highly correlated with the *JGA* because the dialogue states contain constraints for issuing APIs. And, the *DSR* is a more challenging metric compared to *TSR* because the dialogue might contain 2-5 tasks.

**Monolingual vs Bilingual.** Comparing the models that are trained under monolingual and bilingual setting, the latter can leverage more training data and handle tasks in both languages simultaneously without a language identifier. We observe that *mT5* achieves better results in the bilingual settings, while *mBART* performs better with monolingual training. The underlying reason might be the different pre-training strategies of the two mSeq2seq models. *mBART* is pre-trained with language tokens in both the encoder and decoder, but in our bilingual setting, we do not provide any language information. Such a discrepancy does not exist in the *mT5* model, as it is pre-trained without language tokens.

| Models | English (EN) | | | | | Chinese (ZH) | | | | |
|---|---|---|---|---|---|---|---|---|---|---|
| | TSR | DSR | API$_{Acc}$ | BLEU | JGA | TSR | DSR | API$_{Acc}$ | BLEU | JGA |
| *Monolingual* | | | | | | | | | | |
| *MinTL(mBART)* | 56.00 | 33.71 | 57.03 | 35.34 | 67.36 | 56.82 | 29.35 | 71.89 | 20.06 | **72.18** |
| *MinTL(mT5)* | 69.13 | 47.51 | 67.92 | 38.48 | 69.19 | 53.77 | 31.09 | 63.25 | 19.03 | 67.35 |
| *Bilingual* | | | | | | | | | | |
| *MinTL(mBART)* | 42.45 | 17.87 | 65.35 | 28.76 | 69.37 | 40.39 | 16.96 | 65.37 | 5.23 | 69.50 |
| *MinTL(mT5)* | **71.18** | **51.13** | **71.87** | **40.71** | **72.16** | **57.24** | **34.78** | **65.54** | **22.45** | 68.70 |
| *Cross-lingual* | | | | | | | | | | |

| Models | ZH → EN (10%) | | | | | EN → ZH (10%) | | | | |
|---|---|---|---|---|---|---|---|---|---|---|
| | TSR | DSR | API$_{Acc}$ | BLEU | JGA | TSR | DSR | API$_{Acc}$ | BLEU | JGA |
| *MinTL(mBART)* | 1.11 | 0.23 | 0.60 | 3.17 | 4.64 | 0.00 | 0.00 | 0.00 | 0.01 | 2.14 |
| *+ CPT* | 36.19 | 16.06 | 41.51 | 22.50 | 42.84 | 24.64 | 11.96 | 29.04 | 8.29 | 28.57 |
| *+ MLT* | 33.62 | 11.99 | 41.08 | 20.01 | 55.39 | 44.71 | 21.96 | **54.87** | 14.19 | **60.71** |
| *+ TT** | *21.61* | *10.18* | *27.44* | *17.86* | *37.5* | *43.86* | *19.78* | *50.71* | *14.46* | *56.76* |
| *MinTL(mT5)* | 6.78 | 1.36 | 17.75 | 10.35 | 19.86 | 4.16 | 2.20 | 6.67 | 3.30 | 12.63 |
| *+ CPT* | 44.94 | 24.66 | 47.60 | 29.53 | 48.77 | 43.27 | 23.70 | 49.70 | 13.89 | 51.40 |
| *+ MLT* | **56.78** | **33.71** | **56.78** | **32.43** | **58.31** | **49.20** | **27.17** | 50.55 | **14.44** | 55.05 |
| *+ TT** | *56.43* | *34.16* | *57.54* | *31.2* | *58.85* | *47.67* | *26.08* | *50.88* | *14.46* | *54.01* |

Table 2: Dialogue state tracking and end-to-end task completion results in monolingual, bilingual, and cross-lingual settings.

**Cross-lingual.** We observe that it is difficult for the baseline models to converge with minimal training data (10%) due to the complex ontology and diverse user goals. Interestingly, pre-training the mSeq2seq models on the source language improves both DST and task completion performance. Such results indicate the excellent cross-lingual transferability of multilingual language models. Furthermore, the mixed-language training strategy further improves the cross-lingual few shot performance, especially the *JGA*, which suggests that the bilingual knowledge base can facilitate the cross-lingual knowledge transfer in the low resource scenario. Figure 5 shows that, by using 30%-50% target language training data, `mT5+MLT` is able to achieve comparable results to monolingual full training. We also observe that using a translation model does not further improve the results because many name entities are mistakenly translated.

**Limitations and Future Work.** The main limitation of this work is the low number of languages in the corpus due to the difficulty of collecting the knowledge base in languages other than English and Chinese in Hong Kong. In future work, we plan to extend the dataset to more languages including low resource languages in dialogue research (e.g., Cantonese, Indonesian), to better examine the cross-lingual transferability of end-to-end ToD systems. Another limitation is that the M2M data collection might not cover rare and unexpected user behaviours (e.g., non-collaborative dialogues), as dialogue simulators generate the dialogue outlines. We also note that the dialogues in our dataset does not have common features found in spontaneous dialogue [29, 30], such as ellipses, inversions, repairs [31, 32], and split utterances [33], because the utterances in the dataset are collected by paraphrasing (writing). However, we see BiToD as a necessary step for building robust multilingual ToD systems before tackling even more complex scenarios.

## 6 Related Work

**Datasets for End-to-End ToD Modeling.** Many datasets have been proposed in the past to support various assistant scenarios. In English, Wen et al. [34] collected a single domain dataset with a Wizard-of-Oz (Woz) setup, which was latter extended to multi-domain by many follow-up works [17, 18, 35, 36]. Despite its effectiveness, Woz data collection method is expensive since two annotators need to be synchronized to conduct a conversation, and the other set of annotators need to annotate speech-act and dialogue states further. To reduce the annotation overhead (time and cost), Byrne et al. [23] proposed a Machines Talking To Machines (M2M) self-chat annotations schema. Similarly, Rastongi et al.[1, 37] applied M2M to collect a large-scale schema-guided ToD dataset, and Kottur et

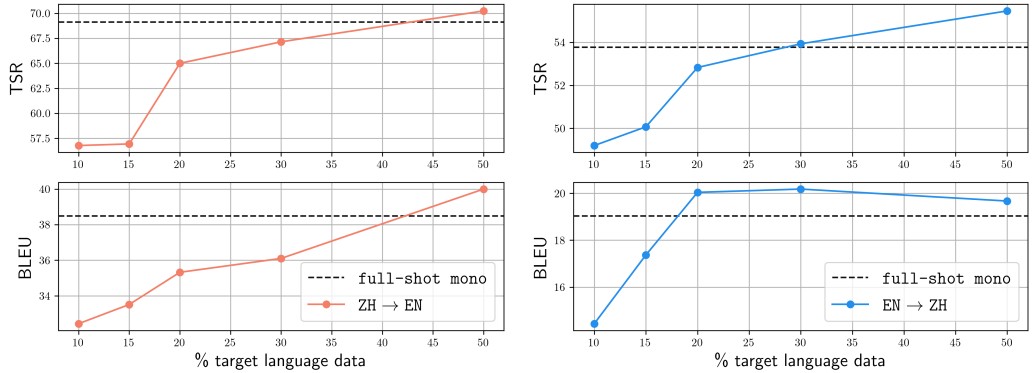

Figure 5: Cross-lingual transfer results (by using `mT5+MLT`) on ZH→EN (left) and EN→ZH (right) with different ratio (%) of target language data. `mT5+MLT` achieves comparable results to monolingual full training by using 30%-50% target language data.

al.[38] extended it to multimodal setting. In languages other than English, only a handful of datasets have been proposed. In Chinese, Zhu et al. [19], and Quan et al.[20] proposed WoZ style datasets, and in German, the WMT 2020 Chat translated the dataset from Byrne et al. [37]. To the best of our knowledge, all the above-mentioned datasets are monolingual, thus making our BiToD dataset unique since it includes a bilingual setting and all the annotations needed for training an end-to-end task-oriented dialogue system. In the chit-chat setting, XPersona [39] has a translation corpus in seven languages, but it is limited in the chit-chat domain. Finally, Razumovskaia et al.[21] made an excellent summarization of the existing corpus for task-oriented dialogue systems, highlighting the need for multilingual benchmarks, like 𝔹𝕚𝕋𝕠𝔻, for evaluating the cross-lingual transferability of end-to-end systems.

**Multilingual Datasets for Spoken Language Understanding (SLU).**    Different from aforementioned datasets focus on dialogue level modeling, there are many SLU datasets [40, 41, 42, 43] have been proposed for utterance level semantic parsing. They serve as effective benchmarks for SLU tasks, such as intent classification, slot filling [44, 45]. In particular, Multi-SLU [46], Multi-ATIS [47], and MTOP [48] are multilingual datasets designed to study cross-lingual transfer learning for SLU. There are several works that applied machine translation models for cross-lingual transfer. Lacalle et al.[49] translated the high resource language dataset to target language for both intent classification and slot filling tasks. Jain et al. [50] leveraged a pre-trained machine translation model to translate entities from source annotated named entity recognition data to the target raw data. In Multi-SLU, Schuster et al. proposed to train a bidirectional neural machine translation model and combine it with an auto-encoder objective for transfer the knowledge from the English data to low resource languages, and this approach has been improved by Liu et al. via a Mixed-Language Training (MLT) strategy [25]. Compared to SLU tasks, which typically involve classification and sequence labeling, the end-to-end modeling task in 𝔹𝕚𝕋𝕠𝔻 involves API query generation and knowledge grounded response generation.

# 7   Conclusion

We present 𝔹𝕚𝕋𝕠𝔻, the first bilingual multi-domain dataset for end-to-end task-oriented dialogue modeling. 𝔹𝕚𝕋𝕠𝔻 contains over 7k multi-domain dialogues (144k utterances) with a large and realistic knowledge base. It serves as an effective benchmark for evaluating bilingual ToD systems and cross-lingual transfer learning approaches. We provide state-of-the-art baselines under three evaluation settings (monolingual, bilingual and cross-lingual). The analysis of our baselines in different settings highlights 1) the effectiveness of training a bilingual ToD system compared to two independent monolingual ToD systems, and 2) the potential of leveraging a bilingual knowledge base and cross-lingual transfer learning to improve the system performance under low resource conditions.

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
