# OpenReview forum: "BiToD: A Bilingual Multi-Domain Dataset For Task-Oriented Dialogue Modeling"
_NeurIPS.cc/2021/Track/Datasets_and_Benchmarks/Round1 — NeurIPS 2021 Datasets and Benchmarks Track (Round 1)_

### Official Review · Reviewer_NRnM · 2021-06-28
**This paper describes a bilingual (chinese-English) multi-domain dataset for end-to-end task oriented dialogue to be used as a benchmark for evaluating ToD systems**

**Rating:** 7
**Confidence:** 4
**Clarity:** The paper is clearly and well-written…

**Strengths:**

- First dataset of its kind
- Sound collection procedure and useful baselines

**Weaknesses:**

- Limited variability compared to natural language dialogue patterns
- Not informed by any previous theoretical work on dialogue in theoretical and computational linguistics before the advent of deep learning in NLP.

**Additional Feedback:**

Some references for the authors w.r.t dialogue work that can be useful in the discussion of dialogue disfluencies, range of dialogue patterns:

Raquel Fernández. Non-Sentential Utterances in Dialogue: Classification, Resolution and Use. PhD thesis, Department of Computer Science, King's College London, University of London, London, UK, 2006.
Howes, C. (2012). Coordination in dialogue: Using compound contributions to join a party. Phd Thesis, Queen Mary University of London.
Gregoromichelaki, E., & Kempson, R. (2016). Joint utterances and the (split-) turn taking puzzle. In Interdisciplinary studies in pragmatics, culture and society (pp. 703-743). Springer, Cham.
Ginzburg J, Fernández R, Schlangen D. Disfluencies as intra-utterance dialogue moves. Semantics and Pragmatics. 2014 Jun 10;7:9-1.
Ginzburg, J., 2012. The interactive stance. Oxford University Press.


**Correctness:**

Dataset is constructed in a sound way and the evaluation methods are appropriate.

**Documentation:**

There is sufficient detail on data collection and organization, availability and maintenance, and ethical and responsible use. The same goes for reproducibility, to the best of my knowledge.

**Relation To Prior Work:**

There is an adequate discussion on similar datasets. What is missing, which is something that most recent papers in the same category seem to miss, is a discussion or rather being informed from earlier literature in dialogue modelling spanning at least fifty years of research that could greatly inform the collection and variability of dialogue patterns found in natural language.

**Summary And Contributions:**

The paper describes a bilingual (chinese-English) multi-domain dataset for end-to-end task oriented dialogue. The paper involves the following contributions:

- First ToD bilingual multi-domain dataset
-  Three baselines for a range of settings (monolingual, bilingual, crosslingual)
- Evidence for using bilingual instead of monolingual ToD systems

The paper is clearly written and easy to follow in general, while the ethical section is also clear and straightforward with no problems whatsoever. This is definitely an important resource for dialogue NLP practicioners. A major weakness however, which is also quite commonly found in other papers in NLP, is the disregard of earlier work in dialogue modelling in linguistics and computational linguistics before the advent of DL in NLP. This is of course not a problem a priori, but because of that, misses key aspects of dialogue that have been extensively been discussed in the literature on dialogue in the last 40 years or so. The most clear example of the latter is that dataset involves cases of clean cut dialogue, namely cases where the abundant disfluencies and idiosyncracies found in natural language, i.e. interjections, split utterances, word repetitions, are not found. This is understandable for datasets that serve a specific practical application, but nevertheless casts doubt into claims on how "natural" such datasets are. In any case, datasets, of which the dataset described in the paper under cosideration is not an exception, involve a fraction of the wealth of patterns found in natural language dialogue. This does not mean that the datasets are not useful. To the contrary: they are extremely useful, but it's good time that these issues start to be discussed in the literature and properly acknowledged.

In general the paper is an important contribution to the community and I could see it being accepted subject to the authors taking into consideration a number of more specific criticisms that are raised below:

- The authors claim that the dataset involves also cases of code-switching. Technically, this is true. However, code-switching is very restricted to the interchange between Chinese and English language names (i.e. of restaurants). This limitation has to be noted.
- Give a clear definition of what constitutes a dialogue
- Give a definiton of what multi-domain is meant to be and what the limitations of those domains are (e.g. how they can be extended to include more domains)
- Discuss the issue of wealth of dialogue patterns found in the dataset and why common features found in spontaneous dialogue (e.g. difluencies, split utterances) are not found in this dataset.

---

> ### Author Response · Authors · 2021-07-09
> **Thank you for your important comments, We are adding more related work in dialogue modelling in linguistics and computational linguistics.**
>
> 1.“However, code-switching is very restricted to the interchange between Chinese and English language names (i.e. of restaurants). This limitation has to be noted.  ”
>
> R: We have noted this limitation in the updated paper (Line 163).
>
> 2. “Give a clear definition of what constitutes a dialogue.”
>
> 3. “Give a definiton of what multi-domain is meant to be and what the limitations of those domains are (e.g. how they can be extended to include more domains) ”
>
> R:  In this paper, we focus on one-to-one conversations that only involve two speakers(e.g., a user and a system). A dialogue is a sequence of utterances , each a single contribution from one speaker to the dialogue[8]. In ToD, there can be additional API calls in system turn when the user requests the system to search about a certain information. In addition to the aforementioned concept, there are several frequently used terms in ToD. Domains are the topics of the current conversation, for example, restaurant domain is about restaurant reservation and metro domain is about taking a metro to somewhere. Multi-domain dialogues are referred as dialogues that involve more than one domain. As shown in Figure 1, the English dialogue involves both restaurant and attraction domains. We have included both definitions in the updated paper (Line 48).
>
> 4. “Discuss the issue of wealth of dialogue patterns found in the dataset and why common features found in spontaneous dialogue (e.g. disfluencies, split utterances) are not found in this dataset. “
>
> 5. “Some references for the authors w.r.t dialogue work that can be useful in the discussion of dialogue disfluencies, range of dialogue patterns”
>
> R: The dialogues in our dataset do not have common features found in spontaneous dialogue[1,2], such as ellipses, inversions, repairs [3], disfluencies [4], and split utterances [5], because the utterances in the dataset are collected by paraphrasing (writing) after the annotators read through the dialogue outlines. We noted this as a common limitation in current text-based ToD datasets [6,7], and we have included all the references and discussed this limitation in our updated paper (line 278).
>
> Thank you for all the feedback. Hope our responses address your concerns. Please  Feel free to let us know if you have any further feedback.
>
> [1] Raquel Fernández. Non-Sentential Utterances in Dialogue: Classification, Resolution and Use. PhD thesis, Department of Computer Science, King's College London, University of London, London, UK, 2006.
>
> [2] Howes, C. (2012). Coordination in dialogue: Using compound contributions to join a party. Phd Thesis, Queen Mary University of London.
>
> [3] Itoh, Toshihiko, et al. "A robust dialogue system with spontaneous speech understanding and cooperative response." Interactive Spoken Dialog Systems: Bringing Speech and NLP Together in Real Applications. 1997.
>
> [4] Ginzburg J, Fernández R, Schlangen D. Disfluencies as intra-utterance dialogue moves. Semantics and Pragmatics. 2014 Jun 10;7:9-1. Ginzburg, J., 2012. The interactive stance. Oxford University Press.
>
> [5] Gregoromichelaki, E., & Kempson, R. (2016). Joint utterances and the (split-) turn taking puzzle. In Interdisciplinary studies in pragmatics, culture and society (pp. 703-743). Springer, Cham.
>
> [6] Rastogi, A., Zang, X., Sunkara, S., Gupta, R., & Khaitan, P. (2020, April). Towards scalable multi-domain conversational agents: The schema-guided dialogue dataset. In Proceedings of the AAAI Conference on Artificial Intelligence (Vol. 34, No. 05, pp. 8689-8696).
>
> [7] Budzianowski, P., Wen, T. H., Tseng, B. H., Casanueva, I., Ultes, S., Ramadan, O., & Gasic, M. (2018). MultiWOZ-A Large-Scale Multi-Domain Wizard-of-Oz Dataset for Task-Oriented Dialogue Modelling. In Proceedings of the 2018 Conference on Empirical Methods in Natural Language Processing (pp. 5016-5026).
>
> [8] Jurafsky, D. (2000). Speech & language processing. Pearson Education India.

---

> > ### Author Response · Authors · 2021-07-15
> > **Thank you for all the suggestions. We are happy to provide any additional information.**
> >
> > Dear reviewer NRnM,
> >
> > Thank you for all the suggestions. As the discussion period is going to end in few hours, we would like to check if you have any further comments based on our updated paper. We are happy to provide any additional information.

---

> ### Comment · Area_Chair_LfTs · 2021-07-19
> **author response**
>
> Dear reviewer, Could you take a look at their response? The reviewer discussion period ends tomorrow, and it'd be really helpful if you can comment whether the author response changed your opinion of the paper, and update the review accordingly. Thanks!

---

### Official Review · Reviewer_YchJ · 2021-07-03
**A new bilingual multi-domain dataset for end-to-end task-oriented dialogue modeling**

**Rating:** 7
**Confidence:** 2
**Correctness:** Yes. The dataset collection process s…
**Clarity:** Overall, I feel it is easy to follow …

**Strengths:**

1. The authors provides detailed explanation about the data collection process, intended data usage,  dataset statistics and maintenance plan.
2. Two sate-of-the-art baseline models are evaluated across three evaluation settings (monolingual, bilingual, and cross-lingual). Analysis show leveraging bilingual data can train a better ToD system for both languages.

**Weaknesses:**

I feel some details/baselines are missing for better understanding about the dataset. I might be wrong about this, as I am not very familiar with dialogue systems.

(1) Is there one-to-one correspondence between the English dialogues and Chinese dialogues. For example, an English dialogue and a Chinese dialogue generated with the same user goal, hence share similar dialogue outline in meaning.

If so, can we obtain a baseline performance based on machine translation. Is it possible to directly transfer a learned model with English training data to Chinese testing data via machine translation?

(2) How many paraphrased dialogues are collected for a dialogue outline?

(3) Upper bound of model performance or human performance.
For all accuracy measure, I suppose the upper bound is 100%. It may be difficult to get human performance to mimic the whole dialogue pipeline. However, for generated response, what kind of BLEU score would suggest the generated response is good.

(4) Cross-domain transferability. As BiToD includes dialogues from different domains, would it be possible to evaluate a dialogue system on cross-domain transferability? For example, train the model with data in one domain and evaluate on data in other domains.

(5) Visualization of model predictions. It would be nice to have a check on the qualities of model generated responses and identify the failure cases under different evaluation settings to highlight the difficulties of this bilingual dataset.

(6) Under the cross-lingual setting, if we keep increase the training data in the other language (e.g., the percentage of English data used for ZH->EN ) , when would the model get comparable performance with using the full English data under monolingual setting.

**Additional Feedback:**


It is great to further extend BiToD to other languages as future work. What about diversifying the knowledge base? For example, current knowledge base are about entities in Hongkong, would it be possible to extend the dataset to other regions (cities or nations)? I am curious about whether a dialogue system trained with data designed for one city can be transferred to another city. I suppose the model should have leaned how to identify the APIs, understand the goal of a dialogue and etc. I would like to hear about the authors' thoughts.


**Post-rebuttal Updates** The authors have addressed my concerns. So I raised my rating to acceptance.

**Documentation:**

Yes. The authors have included details about data maintenance, collection, organization and intended use in appendix. The data and baseline code are included in their provided Github repository.

**Ethics:**

No.

**Relation To Prior Work:**

Yes. In section 6, the authors have summarized prior works on building datasets for end-to-end task-oriented dialogue modeling. The biggest difference between BiToD and prior works is the bilingual nature of BiToD.

The authors have also provided detailed comparison of dataset statistics/features between BiToD and previous datasets in Table 2. In addition to the bilingual nature, BiToD provides several additional features: Deterministic API, Complex User Goal, Mixed-Language Context and Cross-API Entity Carry-Over.

**Summary And Contributions:**

The paper presents a new dataset BiToD, a task-oriented dialogue benchmark in bilingual setting (English and Chinese) across multiple domains (such as: restaurant, attraction).  Experiments conducted on two state-of-the-art baselines demonstrate the need of such bilingual dataset.

---

> ### Author Response · Authors · 2021-07-09
> **Thank you for all the valuable feedback, we have updated our paper accordingly.**
>
> 1. “Is there one-to-one correspondence between the English dialogues and Chinese dialogues? … Is it possible to directly transfer a learned model with English training data to Chinese testing data via machine translation?”
>
> R: There is no one-to-one correspondence between the English dialogues and Chinese dialogues as English and Chinese user goals are sampled independently. However,  we included the translation baselines in the updated paper (results in Table 2, described in Line 244). We found that using a translation model does not further improve the results because many name entities are mistakenly translated.
>
> 2. “How many paraphrased dialogues are collected for a dialogue outline?”
>
> R: We sample around 7000 dialogue outlines, and each outline is only paraphrased once.
>
> 3. “However, for generated response, what kind of BLEU score would suggest the generated response is good?”
>
> R: In the ToD response generation task, BLEU is used to measure the similarity between the generated responses and the human responses. In practice, the responses with 20 or higher BLEU score are considered relatively good responses in ToD.
>
> 4. “As BiToD includes dialogues from different domains, would it be possible to evaluate a dialogue system on cross-domain transferability? ”
>
> R: To the best of our knowledge, there is no existing work on cross-domain end-to-end dialogue task completion. Because models cannot predict a new API call without target domain training data. In our paper, we focused on the multi-linguality aspect of ToDs. However, this research direction is very interesting and it could be explored in future work also using our proposed dataset.
>
> 5. “Visualization of model predictions. It would be nice to have a check on the qualities of model generated responses and identify the failure cases under different evaluation settings to highlight the difficulties of this bilingual dataset.”
>
> R: Thank you for the suggestion. We conduct the error analysis on the three settings and the results are shown in the Appendix Table 5-9. Overall, we found several recurrent errors across different settings such as 1) errors in the generated DST, both for wrong values, addition or missing slots, and wrong API names, leading to wrong responses, and 2) the model mistakenly generated an API rather than responses. We have included this analysis in Line 770.
>
>
> 6. “Under the cross-lingual setting, if we keep increasing the training data in the other language (e.g., the percentage of English data used for ZH->EN ) , when would the model get comparable performance with using the full English data under monolingual setting.”
>
> R. We have added this result in the paper. In our experiments, by only using around 30% target language data, the model is able to achieve comparable results to the full training (as shown in Figure 5).
>
> 7 “What about diversifying the knowledge base? For example, current knowledge bases are about entities in Hongkong, would it be possible to extend the dataset to other regions (cities or nations)? I am curious about whether a dialogue system trained with data designed for one city can be transferred to another city.”
>
> R: It is possible to extend the dataset to other regions. The only constraint would be to map the service data (e.g., restaurants information) in other countries to the same ontology of the existing data.
>
> Thank you for all the feedback. If you find our responses addressed your concerns effectively, please consider increasing the rating.  Feel free to let us know if you have any further comments.

---

> > ### Author Response · Authors · 2021-07-15
> > **Thank you for all the suggestions. We are happy to provide any additional information.**
> >
> > Dear reviewer YchJ,
> >
> > Thank you for all the suggestions. As the discussion period is going to end in few hours, we would like to check if you have any further comments based on our updated paper. We are happy to provide any additional information.

---

> ### Comment · Area_Chair_LfTs · 2021-07-19
> **author response**
>
> Dear reviewer, could you take a look at author'ss response? The reviewer discussion period ends tomorrow, and it'd be really helpful if you can comment whether the author response changed your opinion of the paper, and update the review accordingly. Thanks!

---

### Official Review · Reviewer_sTQQ · 2021-07-03

**Rating:** 5
**Confidence:** 4
**Correctness:** 1. The evaluation methods and experim…
**Clarity:** 1. Yes, the main paper is generally w…

**Strengths:**

1. The dataset will be useful towards better evaluating multilingual dialog systems for the development of robust end-to-end dialog systems for multilingual countries and regions.
2. The dataset and code is already publicly available.
3. The dataset is quite large and diverse (and multilingual) in comparison with existing dialog datasets.
4. The authors have done a good job implementing several classes of baselines and the results look interesting.

**Weaknesses:**

1. There needs to be comparison with other multilingual dialog datasets and models beyond english and chinese, such as
Schuster et al., Cross-Lingual Transfer Learning for Multilingual Task Oriented Dialog. NAACL 2019 https://arxiv.org/abs/1810.13327
2. There could be more discussion regarding the annotation process since there may be possible biases resulting from the annotation process given that native english or chinese speakers follow different distributions, and may have different ways of responding and completing the dialog task. More discussion regarding checks for the accuracy of annotations would also help the paper.
3. There should be some references to machine translation literature and how using pretrained machine translation models for cross-lingual transfer can help.

**Additional Feedback:**

none.

**Documentation:**

1. There could be more discussion regarding the annotation process since there may be possible biases resulting from the annotation process given that native english or chinese speakers follow different distributions, and may have different ways of responding and completing the dialog task.
2. Perhaps also more discussion regarding the accuracy of annotations could help.

**Ethics:**

1. The authors write that 'Our dataset neither introduces any social/ethical, since we generate data with dialogue simulator and humanly paraphrase the utterances nor amplifies any bias'. Not sure about this, since human paraphrasing could introduce social and ethical biases according to the annotator. More details here could help, including possibly checks to ensure that the dataset does not contain social biases.

**Relation To Prior Work:**

1. There could be more references and comparison to multilingual dialog datasets and models beyond english and chinese, such as these which I found on a cursory google search:
Schuster et al., Cross-Lingual Transfer Learning for Multilingual Task Oriented Dialog. NAACL 2019, https://arxiv.org/abs/1810.13327
Lacalle et al., Building a Task-oriented Dialog System for languages with no training data: the Case for Basque. LREC 2020, https://aclanthology.org/2020.lrec-1.340.pdf
2. Some references to machine translation literature would be relevant too, especially when applied in a dialog setting.

**Summary And Contributions:**

This paper introduces a bilingual multi-domain dataset for end-to-end task-oriented dialogue modeling. Their dataset is large, diverse, and publicly available with over 7k multi-domain dialogues (144k utterances) together with a large and realistic bilingual knowledge base. The authors evaluate several state-of-the-art bilingual ToD baselines and show the importance of training bilingual dialog models over monolingual ones, which their dataset can help with. The authors also explore several cross-lingual transfer baselines in enabling ToD under low resource languages.

---

> ### Author Response · Authors · 2021-07-09
> **Thanks for all the insightful comments, we have updated our paper accordingly.**
>
> 1. “There could be more references and comparison to multilingual dialog datasets and models beyond english and chinese.”
>
> R: We have carefully read the two papers [1, 2] and summarize the comparison below:
>
> Both papers [1, 2] used the Multilingual dataset from [1], which is proposed for Spoken Language Understanding (SLU), and it provides data only in utterance level (40k independent utterances), instead of dialogue level. Therefore, they usually serve as testbeds for SLU tasks, such as intent classification and slot filling. There are some other datasets focused on multilingual SLU, such as Multi-ATIS [3] and Multi-TOP [4].
>
> Different from these datasets, BiToD focuses on dialogue level tasks, such as dialogue state tracking (DST) and end-to-end task completion (E2E), which involve dialogue level understanding and knowledge base interaction. Therefore, the datasets/models here are not directly comparable to SLU datasets/models.
> We have cited all these papers and included this discussion in our updated paper (Line 303).
>
> 2. “There could be more discussion regarding the annotation process since there may be possible biases resulting from the annotation process. … More discussion regarding checks for the accuracy of annotations would also help the paper.”
>
> R: The only data annotation step that might introduce the biases is the dialogue paraphrase step. We have carefully studied the potential bias of the paraphrased utterances. In the dialogue outlines, we always sample user names from 100 most common male names and female names uniformly. However, after the dialogue paraphrase, we found that, out of 144,000 utterances, the assistant call the user “Sir” 73 times and “Madam” 8 times in the English dialogues, while in Chinese dialogues, the assistant call the user “先生(Sir)” 162 times and “女士(Madam)” 2 times. Nevertheless, this social bias issue has been addressed by replacing these with gender-neutral terms (Sir/Madam). We have included this discussion in the Ethics Statements (Line 522).
>
> For the annotation checking, we had implemented an automatic checker for all the essential entities to ensure there is no important information missing. After the dialogue paraphrasing, we randomly sampled around 3000 dialogues, and asked different sets of annotators to check the quality of paraphrased dialogues (in terms of naturalness of the language and dialogue flow). We found around 103 (3.43%) low-quality dialogues, and all of them have been annotated. We have added this discussion in the updated paper (Line 136).
>
> 3. “why the authors chose english and chinese as their 2 languages in the datasets given their very different vocabularies and language similarities.”
>
> R:  English and Chinese are indeed very different. Together they provide a point of reference for cross-lingual transfer methods. Another reason is that we collected our knowledge base in Hong Kong where most of the services are only in English and Chinese, the two official languages of Hong Kong. Nevertheless, we found that cross-lingual transferring is quite effective as the multilingual model like mT5 has learned good alignment among languages. In our experiments, by only using around 30% target language data, the model is able to achieve comparable results to the full training (as shown in Figure 5).
>
> 4. “There should be some references to machine translation literature and how using pretrained machine translation models for cross-lingual transfer can help”
>
> R: we have included a discussion of using pretrained machine translation models for cross-lingual transfer [1, 6] in the updated paper (Line 308). In addition, we have added a new baseline by using the recent state-of-the-art translation model M2M-100 [7] (Line 244). The results have been included in Table 2. We found that using a translation model does not further improve the results because many name entities are mistakenly translated.
>
> Thank you for all the feedback. If you find our responses addressed your concerns effectively, please consider increasing the rating. Feel free to let us know if you have any further comments.
>
> [1] Cross-Lingual Transfer Learning for Multilingual Task Oriented Dialog. NAACL 2019
>
> [2]  Building a Task-oriented Dialog System for languages with no training data: the Case for Basque. LREC 2020
>
> [3] (Almost) Zero-Shot Cross-Lingual Spoken Language Understanding. ICASSP 2018
>
> [4] MTOP: A Comprehensive Multilingual Task-Oriented Semantic Parsing Benchmark. EACL 2021
>
> [5] Multilingual seq2seq training with similarity loss for cross-lingual document classification. RepL4NLP. 2018
>
> [6] Entity Projection via Machine-Translation for Cross-Lingual NER. EMNLP. 2019
>
> [7] Beyond English-Centric Multilingual Machine Translation. JMLR 2021 (https://www.jmlr.org/papers/volume22/20-1307/20-1307.pdf)

---

> > ### Author Response · Authors · 2021-07-15
> > **Thank you for all the suggestions. We are happy to provide any additional information.**
> >
> > Dear reviewer sTQQ,
> >
> > Thank you for all the suggestions. As the discussion period is going to end in few hours, we would like to check if you have any further comments based on our updated paper. We are happy to provide any additional information.

---

> > ### Comment · Area_Chair_LfTs · 2021-07-19
> > **author response:**
> >
> > Dear reviewer, as authors have commented out -- Could you take a look at their response? The reviewer discussion period ends tomorrow, and it'd be really helpful if you can comment whether the author response changed your opinion of the paper, and update the review accordingly. Thanks!

---

### Author Response · Authors · 2021-07-10
**Thank you for all the important comments, we have incorporated all the suggestions in our updated paper.**

We would like to thank the reviewers for their suggestions. We have made the following updates in the paper:

1. A clear definition of what constitutes a dialogue and a definition of what multi-domain dialogue. (Reviewer NRnM)

2. Noted the limitation of code-switching cases in our dataset.  (Reviewer NRnM)

3. Applied the state-of-the-art machine translation model in our cross-lingual setting. (Reviewer sTQQ and YchJ)

4. Discussed all the multilingual SLU datasets and models beyond English and Chinese. (Reviewer sTQQ)

5. Discussed the possible biases during dialogue paraphrase and how we address this issue. (Reviewer sTQQ)

6. Conducted error analysis of baselines under different settings. (Reviewer YchJ)

7. Gradually increase the training data in the target language until the cross-lingual performance matched the monolingual full training. (Reviewer YchJ)

8. Discussed the issue of the wealth of dialogue patterns found in the dataset and why common features found in spontaneous dialogue (e.g. ellipses, split utterances) are not found in this dataset.   (Reviewer NRnM)

All the updates are highlighted in red text.

---

> ### Author Response · Authors · 2021-07-14
> **Any further comments are welcome.**
>
> Dear reviewers,
>
> As the discussion period is ending soon, we would like to know if you have any further comments based on our updated paper. We are happy to answer any questions you have.

---

### Decision · Program_Chairs · 2021-07-26

**Decision:**

Accept

**Comment:**

This dataset introduces the first large-scale (144K utterances over 7K dialogues) bilingual multi-domain dataset for end-to-end task-oriented dialogue modeling. All reviewers agreed the experiments and baselines are sound and clearly presented. The experiments are done in monolingual, bilingual, and cross-lingual settings. The authors distinguish their dataset with existing datasets, which either only focuses on English or does utterance-level study instead of dialogue-level study for different languages.

The authors have improved the paper based on the reviews, adding additional baseline (translate-test), results (using more target language in cross-lingual settings), and discussions (about limitations of the work).

Lastly, as the authors bring up Hong Kong as the reason they investigate English and "Chinese", it would be better to be more specific which Chinese they are focusing on (Mandarin or Cantonese), as Cantonese is (still) the majority Chinese language in HK. If they are not making such distinctions during data collection, this should be clarified in the paper. Lastly, it's still unanswered who annotated this dataset -- were they native speakers of the annotated languages. Please add these details to revised paper.